# A RT-qPCR system using a degenerate probe for specific identification and differentiation of SARS-CoV-2 Omicron (B.1.1.529) variants of concern

**Randi Jessen[1], Line Nielsen[1], Nicolai Balle Larsen[1], Arieh Sierra Cohen[1], Vithiagaran Gunalan[2], Ellinor Marving[2], Alonzo Alfaro-Núñez[3], Charlotta Polacek[2], The Danish COVID-19 Genome Consortium (DCGC)[¶], Anders Fomsgaard[2], Katja Spiess[2]***

**1** Test Center Denmark, Statens Serum Institut, Copenhagen, Denmark, **2** Department of Virus and Microbiological Special Diagnostics, Statens Serum Institut, Copenhagen, Denmark, **3** Department of Clinical Biochemistry, Naestved Hospital, Naestved, Denmark

¶ The complete membership of the author group can be found in the acknowledgments. https://www.covid19genomics.dk.

* ktsp@ssi.dk

**Data Availability Statement:** All relevant data are available from the GISAID data base. The GISAID

## Abstract

Fast surveillance strategies are needed to control the spread of new emerging SARS-CoV-2 variants and gain time for evaluation of their pathogenic potential. This was essential for the Omicron variant (B.1.1.529) that replaced the Delta variant (B.1.617.2) and is currently the dominant SARS-CoV-2 variant circulating worldwide. RT-qPCR strategies complement whole genome sequencing, especially in resource lean countries, but mutations in the targeting primer and probe sequences of new emerging variants can lead to a failure of the existing RT-qPCRs. Here, we introduced an RT-qPCR platform for detecting the Delta- and the Omicron variant simultaneously using a degenerate probe targeting the key ΔH69/V70 mutation in the spike protein. By inclusion of the L452R mutation into the RT-qPCR platform, we could detect not only the Delta and the Omicron variants, but also the Omicron sub-lineages BA.1, BA.2 and BA.4/BA.5. The RT-qPCR platform was validated in small- and large-scale. It can easily be incorporated for continued monitoring of Omicron sub-lineages, and offers a fast adaption strategy of existing RT-qPCRs to detect new emerging SARS-CoV-2 variants using degenerate probes.

## Introduction

The World Health Organization designated the SARS-CoV-2 variant B.1.1.529 as a variant of concern (VOC) with the name 'Omicron' on November 26th, 2021 [1]. Omicron, first identified in South Africa and Botswana in November 2021, has spread rapidly worldwide and is the current dominant strain in at least 166 nations worldwide, replacing the Delta variant [2]. Since its designation as a VOC, Omicron has continued to evolve, leading to descendent lineages, termed BA.1-5, with different genetic constellations of mutations [3].

ID numbers are provided in the Supporting Information files.

**Funding:** The authors received no specific funding for this work.

**Competing interests:** The authors have declardd that no competing interest exist.

The Omicron variant is exceptional for carrying a large number of mutations not seen in combination before, including more than 40 mutations in the spike protein [4, 5]. The spike protein is the main target of antibodies generated by infection and of many vaccines widely administered, and these mutations are therefore associated with increased transmissibility [6] and immune escape [7, 8].

Initially, the BA.1 sub-variant was the dominant circulating version of Omicron. However, since the beginning of 2022, the proportion of BA.2, confirmed by whole genome sequencing (WGS), has increased dramatically, making BA.2 the dominant SARS-CoV-2 virus globally [2]. BA.1 and BA.2 differ by 19 signature mutations located in the N-terminal and in the receptor-binding domain (RBD) of the spike gene, which makes them as different as some other major variants, *e.g.* the diversity between the original Wuhan and the Alpha variant in the spike gene is less than between BA.1 and BA.2 [4, 5]. One notable difference between BA.1 and BA.2 is that BA.2 lacks the characteristic S-gene target failure (SGTF)-causing deletion, ΔH69/V70. This key mutation, a 2-amino acid deletion, has appeared in multiple SARS-CoV-2 variants, including the Alpha variant (B.1.1.7) [9]. BA.4 and BA.5 have an identical BA.2-like constellation of mutations in the spike protein, but carry additional signature mutations including S: ΔH69/V70 and S: L452R. The differences between BA.4 and BA.5 instead lie outside the spike gene. The first cases of BA.4 and BA.5 were reported from the Republic of South Africa in January 2022 [2] and have subsequently been found in at least 60 countries [2]. In Denmark, 1,091 patient samples infected with BA.4 and 6,685 infected with BA.5 have been confirmed by WGS per 22$^{nd}$ of June, 2022 [10]. BA.4 and BA.5 display higher transmission than other Omicron sub-variants and partially evade immunity from natural infections or vaccines to a higher extent than BA.1 and BA.2, enabling BA.4 and BA.5 to infect people who were immune to prior sub-lineages of Omicron and other variants [11]. As Omicron continuously evolves under immune pressure, it may develop mutations that would undermine most broadly-neutralizing therapeutic antibodies.

Denmark has one of the highest SARS-CoV-2 Reverse Transcription quantitative Real-time PCR (RT-qPCR) testing and WGS capacities in the world [12]. As part of this, a flexible RT-qPCR platform termed Variant-PCR was developed for identification and monitoring of circulating SARS-CoV-2 variants of concern (VOCs) in near real-time, enabling rapid decision-making about testing, contact tracing and isolation [13]. The Variant-PCR platform utilizes signature mutations or combinations hereof that allow differentiation of the most critical VOCs by detection of four signature mutations in the SARS-CoV-2 spike protein: ΔH69/V70, L452R, E484K and N501Y [13].

We designed a degenerate probe for the detection of the ΔH69/V70 key mutation in both Alpha and Omicron variants. By combining additional results from L452R RT-qPCR, we were able to distinguish between the BA.1 and BA.2 sub-lineages, the Delta variant, and furthermore detect BA.4 and BA.5 as they emerged, although differentiation between BA.4 and BA.5 was not possible as they share the same spike protein mutations. Although BA.4 emerged before BA.5 in Denmark, BA.5 is now the dominant SARS-CoV-2 variant [10]. Validation was performed in small-scale and performance was confirmed in large-scale diagnostic settings after adaptation of the Variant-PCR to include the degenerate ΔH69V/70 probe, using paired WGS consensus genomes.

This is the first RT-qPCR platform reported for simultaneous detection and discrimination of Omicron sub-variants. Furthermore, this study proves that a flexible platform such as the Variant-PCR platform can be adapted quickly and easily with a degenerate probe to monitor and differentiate existing and upcoming SARS-CoV-2 variants as part of a national testing program to help policy-makers to make public health decisions.

## Materials and methods

### Ethics

Exemption for review by the ethical committee system was given by the Committee on Biomedical Research Ethics–Capital region in accordance with Danish law on assay development projects. 'The study did not require informed consent. The Statens Serum institute has permission to store excess biological material for surveillance and patients can elect to have their samples excluded for this purpose.

### Samples and controls

TWIST Synthetic SARS-CoV-2 RNA control (MT103907 England/205041766/2020) was purchased from TWIST bioscience in a known concentration (copies/μl) and used as PCR standard Alpha (B.1.17) in a seven-step 1:10 dilution series. The average Ct-values and standard deviations were calculated based on biological duplicates.

Samples from patients with WGS-confirmed SARS-CoV-2 WT, Alpha variant B.1.1.7, Delta variant B.1.617.2 and Omicron variant B.1.529 were obtained from the Danish National Biobank to verify whether the RT-qPCR assays performs with all circulating VOCs [14]. In short, throat swabs were collected and tested SARS-CoV-2 positive by community testing facilities (Test Center Denmark), which form part of the Danish national testing program [12]. Primary diagnosis of SARS-CoV-2 infection was made by E-Sarbeco RT-qPCR [15]. The identities of the lineages BA.1, BA.2, BA.4 and BA.5 were confirmed by WGS generated by the Danish COVID-19 Genome Consortium (DCGC) and consensus genomes were deposited in GISAID (gisaid.org).

Positive controls for large-scale RT-qPCR consisted of heat in-activated (60˚C for 45 minutes) Danish virus isolates of Alpha variant B.1.1.7 (SARS-CoV-2/Hu/DK/SSI-H14, NCBI reference sequence ON784346) and Delta variant B.1.617.2 (SARS-CoV-2/Hu/DK/SSI-H11, NCBI reference sequence OM444216) propagated in Vero E6 cells, covering the key mutations (ΔH69/V70 and L452R) present in the Alpha variant B.1.1.7, Delta variant B.1.617.2 and Omicron variant B.1.1.529. Virus isolation, cultivation and dilution was performed as previously described [13]. Sterile 1x PBS pH 7.2 (Gibco) was used as negative control. Positive and negative controls were run in parallel with selected clinical samples throughout RNA isolation and RT-qPCR as described [13].

### RNA extraction

For small-scale experiments, viral genomes from throat swabs collected in 1xPBS (700 μL) were isolated using MagNa Pure 96 nucleic acid extraction system (Roche) with reagents from the MagNa Pure 96 DNA and Viral NA Small Volume kit with 200 μL sample in 1xPBS as input and 100 μL elution. As positive control material, supernatant from SARS-CoV-2 infected cells (120 μL) were mixed with 120 μL of MagNA Pure Lysis Buffer (Roche) followed by extraction as small-scale SARS-CoV-2 patient samples. Positive control RNA was stored at -80˚C until use. For large-scale SARS-CoV-2 patient screening (Variant-PCR), viral RNA was extracted using Beckman Coulter RNAdvance Viral Reagent kit with 200 μl sample in 1xPBS as input and 50 μl Nuclease-free water for elution on Beckman Coulter Biomek i7 automated workstations.

### RT-qPCR assays

The primers and probes listed in Table 1 were synthesized and HPLC-purified by Biosearch Technologies, Denmark. To perform allelic discrimination analysis of the spike gene, two

**Table 1. Primers and probe sequences and relative concentration used in RT-qPCRs.**

| Target | Name | Oligo | Volume (µL) * | Ref |
|---|---|---|---|---|
| S: Δ69/70 | SARS-CoV-2_Δ69/70 F | ACATTCAACTCAGGACTTGTTCT | 0.1 | [13] |
| | SARS-CoV-2_Δ69/70 R | TCATTAAATGGTAGGACAGGGTT | 0.1 | [13] |
| | SARS-CoV-2_Δ69/70 P | HEX-TTCCATGCTATCTCTGGGACCA-BHQ2 | 0.05 | [13] |
| | SARS-CoV-2_Δ69/70 P_Wob | HEX-TTCCATG**Y**TATCTCTGGGACCA-BHQ1 | 0.05 | |
| | SARS_CoV-2_WT_P3 | Texas Red-TACATGTCTCTGGGACCAAT-BHQ2 | 0.05 | [13] |
| S: L452R | SARS-CoV-2_L452R F | CAGGCTGCGTTATAGCTTGGA | 0.1 | [13] |
| | SARS-CoV-2_L452R R | CCGGCCTGATAGATTTCAGT | 0.1 | [13] |
| | SARS-CoV-2_452R_mutant_BHQ+ P | HEX-TATAATTACCGGTATAGATTGTT-BHQ1 | 0.05 | [13] |
| | SARS-CoV-2_L452_WT BHQ+ P | Cal Fluor Red 610-TATAATTACCTGTATAGATTGTT-BHQ1 | 0.05 | [13] |
| E-gene | E_Sarbeco_F | ACAGGTACGTTAATAGTTAATAGCGT | 0.1 | [15] |
| | E_Sarbeco_R | ATATTGCAGCAGTACGCACACA | 0.1 | [15] |
| | E_Sarbeco_P1 | FAM-ACACTAGCCATCCTTACTGCGCTTCG-BHQ1 | 0.05 | [15] |

Mutation specific nucleotides are highlighted in bold.

*Concentration of all primer and probes is 100 µM.

probes were designed for each key mutation: one detecting the wildtype nucleotide sequence and one detecting the mutation, based on sequence alignments using Geneious Prime 2021.0 [13]. The RT-qPCR assay targeting the E gene (E-Sarbeco) was used as unspecific control to determine the presence of the SARS-CoV-2 and estimate viral load [15].

The reaction mixtures for the RT-qPCR experiments were prepared as follows:

For each 25 µL reaction in a 96-well format used in small-scale experiment, 5 µL extracted RNA was added to a 20 µL reaction mix containing 12.5 µL Luna® Universal Probe One-step RT-qPCR reaction buffer (New England Biolabs Inc), 1.25 µL Luna® Warmstart RT Enzyme mix, primers and probes (100 µM, see volumes in Table 1), brought up to 20µL with nuclease-free water.

For each 15 µL reaction in a 384-well format used for large-scale screening, 5 µL extracted RNA was added to 10 µL reaction mix containing 7.5 µL Luna® Universal Probe One-step RT-qPCR reaction buffer (New England Biolabs Inc), 0.75 µL Luna® Warmstart RT Enzyme mix, primers and probes (100 µM, see volumes in Table 1) and nuclease-free water up to 10 µL. The 384-well mastermix plates were arranged in a quadratic pattern enabling analysis of each patient sample with four different mastermixes in parallel, using first quadrate to detect ΔH69/V70, the second to detect L452R, the third to detect E-Sarbeco and the final well was not in use. This allowed for easy transfer of purified RNA from a 96-well format used during extraction to a 384-well mastermix plate for PCR (*e.g.* isolated RNA in A1 in the 96-well template plate was transferred to A1, A2, B1 and B2 of the 384-mastermix plate). The PCR program consists of reverse transcription at 55˚C for 10 min., initial denaturation at 95˚C for 3 min., followed by 45 cycles of denaturation and annealing/extension at 95˚C for 15 sec., and 58˚C for 30 sec., respectively, regardless of PCR plate format.

All RT-qPCR reactions were performed in a calibrated Bio-Rad CFX real-time PCR instrument. The raw data was analyzed with the Bio-Rad CFX Maestro Software using a predefined threshold cut-off value of 100 RFU as a quality step, in case one of the probes in the allelic discrimination pair failed. Recorded Ct-values and end-RFU were calculated in Maestro version 5.2.8.222 and exported for further data analysis. For the mutations ΔH69/V70 and L452R, detection was based on allelic discrimination, where the end-RFU values were utilized to determine the presence of a mutation. A sample is considered positive based on the following

criteria: Ct-values between 10–38 and end-RFU > 200 at Ct = 45. Validation was performed comparing RT-qPCR data to nucleotide sequences corresponding to the *S* gene of consensus genomes from WGS as described [13].

### Whole genome sequencing

Presence of signature mutations was confirmed by Whole genome sequences generated by The Danish COVID-19 Genome Consortium (DCGC) from PCR-positive samples as described [13].

## Results

### A degenerate ΔH69/V70 probe is specific for Omicron detection

We evaluated the analytical performance characteristics of the existing ΔH69/V70 RT-qPCR included in our Variant-PCR [13] for the qualitative detection of Omicron in clinical samples. SARS-CoV-2 positive clinical samples containing Omicron (B.1.529), Alpha (B.1.1.7) were used as positive control, and Wuhan (SARS-CoV-2 WT), Delta (B.1.617.2) and negative SARS-CoV-2 patient samples were used as negative controls, for the ΔH69/V70 RT-qPCR (Fig 1A). The presence of SARS-CoV-2 was confirmed in positive samples by E-Sarbeco RT-qPCR [15] multiplexed with the ΔH69/V70 RT-qPCR (Fig 1B).

The original ΔH69/V70 RT-qPCR (probe design based on the SARS-CoV-2 Alpha variant sequence) was able to detect the ΔH69/V70 mutation in Omicron confirmed samples (Fig 1A). However, the end fluorescence intensity was reduced about five times in Omicron BA.1 (265 RFU) compared to samples with SARS-CoV-2 Alpha (1407 RFU). Sequence analysis revealed a single C-to-T substitution at nucleotide position 21696 in the Omicron BA.1 sequence, resulting in a mismatch in the middle of ΔH69/V70 probe binding region, which led to the observed reduction of the fluorescence intensity.

In order to detect the ΔH69/V70 mutation in all known SARS-CoV-2 variants, we designed a degenerate ΔH69/V70 probe (SARS-CoV-2_Δ69/70 P_Wob, Table 1). Replacing only the original probe targeting ΔH69/V70 with the degenerate probe led to almost a full recovery of the end fluorescence intensity (Fig 2A) in both SARS-CoV-2 variants (Alpha and Omicron) (Fig 2A). Specificity of the degenerate probe was confirmed by the lack of signal from SARS--CoV-2 variants lacking the H69/V70 deletion (SARS-CoV-2 WT, Delta, Omicron BA.2), as

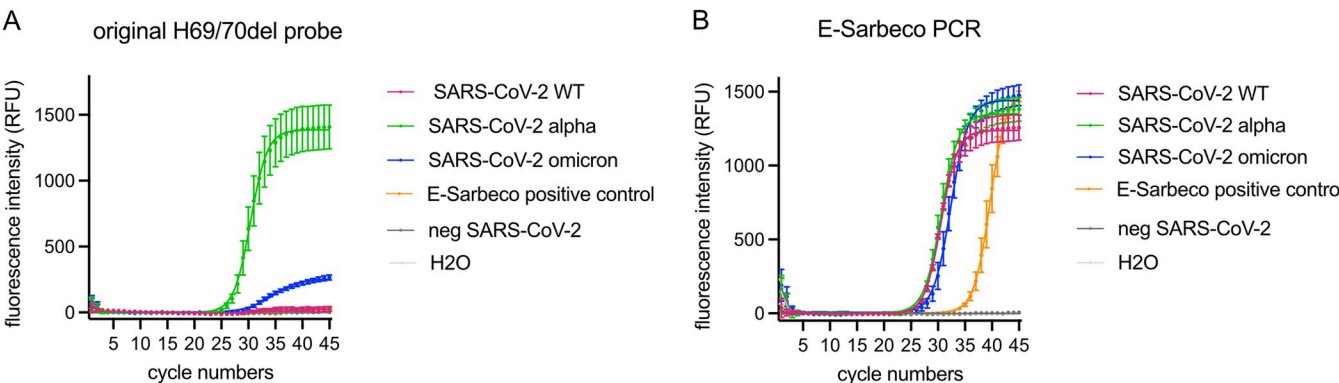

**Fig 1. Detection of SARS-CoV-2 variants with the original ΔH69/V70 probe based on the SARS-CoV-2 alpha variant sequence.** (A) Detection of SARS-CoV-2 variants with the original probe, including patient samples confirmed as SARS-CoV-2 Alpha variant as positive control and SARS-CoV-2 negative patient samples and a water as negative controls. (B) Samples were tested as multiplex RT-qPCRs; E-Sarbeco in parallel with the ΔH69/V70 RT-qPCR. Error bars in (A-B) indicate SEM for three biological replicates.

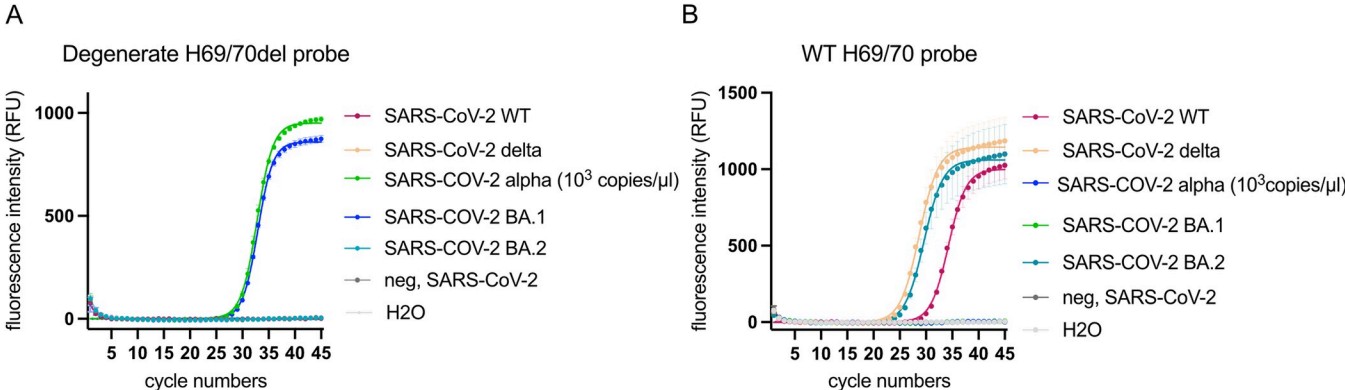

**Fig 2. Detection of SARS-CoV-2 variants with a degenerate ΔH69/V70 probe based on the SARS-CoV-2 BA.1 sequence.** (A) Detection of SARS-CoV-2 variants with a degenerate probe targeting the ΔH69/V70. The SARS-CoV-2 Alpha- and BA.1 variants were included as positive controls, and SARS-CoV-2 negative patient samples and a water as negative controls. (B) Detection of SARS-CoV-2 variants with the WT probe. SARS-CoV-2 WT, Delta and BA.2 variants were included as positive controls and SARS-CoV-2 negative patient samples and a water as negative controls.

well as for the negative controls (Fig 2A). In order to discriminate between BA.1 and BA.2 sub-variants, we included a WT probe in the ΔH69/V70 RT-qPCR. All SARS-CoV-2 variants lacking the ΔH69/V70 (WT, Delta, Omicron BA.2) were detected correctly by the H69/V70 WT probe (Fig 2B).

Moreover, the fluorescence intensity was moderately reduced for the degenerate probe compared to the original probe targeting the ΔH69/V70 present in the SARS-CoV-2 Alpha variant (TWIST control) (S1A Fig). However, the sensitivity of the ΔH69/V70 RT-qPCR was unchanged including the degenerate ΔH69/V70 probe instead of the original probe, as shown by a dilution row of the TWIST standard (Alpha variant) (Table 2; S1B Fig). The sensitivity of the original primer and probe of the ΔH69/V70 RT-qPCR was determined in a previous study [13].

**Table 2. Threshold values (Ct) reported for the TWIST synthetic SARS-CoV-2 RNA quality control assayed with the original and degenerate ΔH69/70 probe in RT-qPCR.**

| SARS-CoV-2 lineage | Dilution factor | Original | Degenerate |
|---|---|---|---|
| | | ΔH69/V70 probe | ΔH69/V70 probe |
| | | (Ct value ± s.d.) | (Ct value ± s.d.) |
| Alpha | $1 \times 10^{-3}$ | 27.5±0.2 | 27.6±0.6 |
| B.1.1.7 | | | |
| Alpha | $1 \times 10^{-4}$ | 31.2±0.3 | 31.7±0.2 |
| B.1.1.7 | | | |
| Alpha | $1 \times 10^{-5}$ | 34.4±0.4 | 35.4±0.007 |
| B.1.1.7 | | | |
| Alpha | $1 \times 10^{-6}$ | 37.2±0.4 | 38.3±0.08 |
| B.1.1.7 | | | |
| Alpha | $1 \times 10^{-7}$ | >45 | > 38.6 |
| B.1.1.7 | | | |
| NTC | 0 | >45 | >45 |

Values reported with a standard deviation are average of duplicates. Threshold value reported as "> 45" indicates no target detection, value reported as ">" indicates the lowest Ct value measured with one replicate > 45. NTC: No template control.

**Table 3. SARS-CoV-2 variants and sub-lineages detected by the combination of ΔH69/V70 and L452R RT-qPCRs.**

| ΔH69/V70 RT-qPCR<br>L452R RT-qPCR | [WT]<br>probe | ΔH69/V70<br>probe |
|---|---|---|
| [WT] probe | Omicron BA.2 | Omicron BA.1 |
| 452R probe | Delta | Omicron BA.4/BA5 |

## Large-scale performance confirmation of allelic discrimination RT-qPCR with degenerate ΔH69/V70 probe

The ΔH69/V70-WT RT-qPCR with the degenerate probe was validated on 745 RT-PCR-positive SARS-CoV-2 patient samples collected from the 3rd to 16th of January 2022 as part of the Danish national testing program. This period was selected since Omicron BA.1 and BA.2 sub-lineages were present in approximately equal proportions [10].

The H69/V70 deletion is not present in the SARS-CoV-2 variants Delta and Omicron BA.2; therefore the L452R RT-qPCR was included into the RT-qPCR large-scale screening platform [13, 16]. Running two RT-qPCRs, each directed against one specific key mutation, *e.g.* ΔH69/V70 and L452R, made it possible to detect four different variants (Table 3); WT at ΔH69/V70 without L452R mutation (characteristic for Omicron BA.2), WT at ΔH69/V70 with L452R mutation (characteristic for the Delta VOC), ΔH69/V70 without L452R mutation (characteristic for Omicron BA.1) and finally, ΔH69/V70 with L452R mutation (characteristic of BA.4 and BA.5 sub-lineages).

Validation was performed by pairwise comparison of results from the Variant-PCR with WGS using the WGS consensus genome from the same samples as the reference standard. We obtained valid RT-qPCR results and a consensus genome sequence from each of the 745 selected SARS-CoV-2 PCR-positive patient samples. In total, 339 patient samples were characterized as BA.1 infections while 399 samples was identified as BA.2 by WGS analysis, showing a nearly even distribution between BA.1 and BA.2 genomes. The Delta genome was found in the last seven patient samples.

The ΔH69/V70 mutation was detected by the degenerate ΔH69/V70 probe in 339 patient samples, which correlated 100% with Omicron BA.1 WGS confirmed samples (Fig 3).

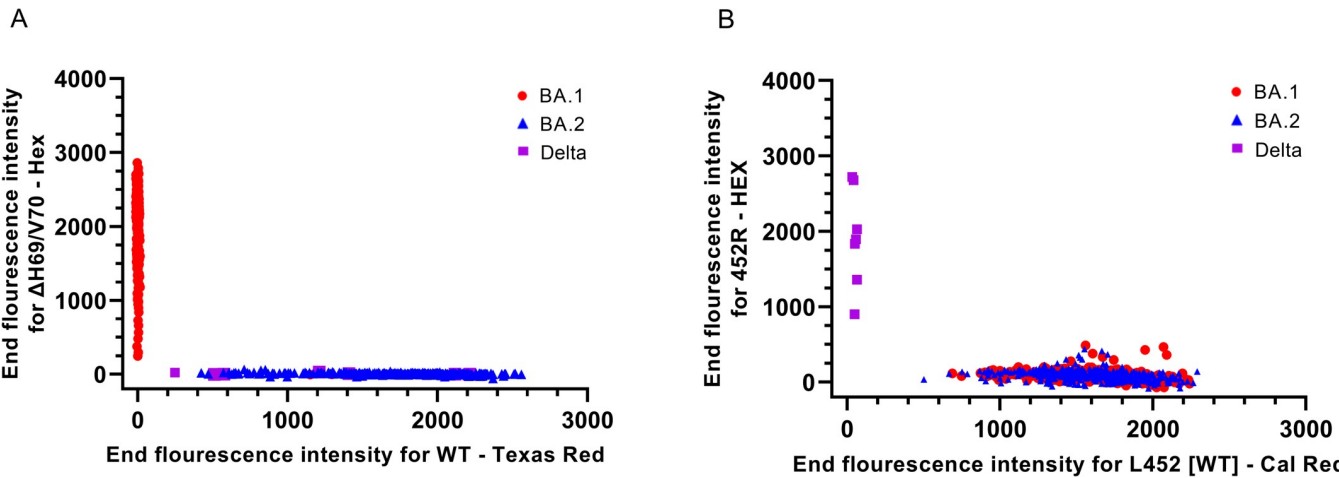

**Fig 3. Detection of ΔH69/V70 and L452R mutations in BA.1 and BA.2.** (A) Allelic discrimination analysis to differentiate between the H69/V70 deletion and the WT sequence. (B) Allelic discrimination analysis to differentiate between the 452R mutation and the L452 WT sequence.

Furthermore, 406 SARS-CoV-2 positive clinical samples had a WT signal being either of Delta or Omicron BA.2 origin. Out of these 406 samples, seven were also found positive for the 452R mutation, characteristic for the Delta variant, which fully corresponded with the WGS results. The L452-WT sequence was detected in 738 out of the 745 patient samples corresponding to the total consensus sequences identified as Omicron by WGS, regardless of BA.1 and BA.2 sub-lineage identity. Thus, all samples correlated between the degenerate ΔH69/V70-WT RT-qPCR and WGS, demonstrating high specificity (>99.9%) and high sensitivity (>99.9%) in large-scale as well as small-scale settings.

In summary, we developed a RT-qPCR system for rapid differentiation of Omicron BA.1 and BA.2 in large-scale, validated by comparison of RT-qPCR and WGS data from 745 patient samples analyzed in parallel.

## Large-scale performance confirmation of BA.4 and BA.5 Omicron variants

The Danish national SARS-CoV-2 genomic surveillance system identified the first case of BA.4 in a sample from the 2nd of March, 2022 based on WGS. On the 15th of April, 2022, the first case of BA.5 case was detected. At this point in time, BA.2 accounted for more than 99% of all sequenced variants in Denmark [10]. The prevalence of BA.4 increased at a linear rate from 0.1% to 4.7% during the period from the 11th of April to 1st of June, 2022 with 327 identified BA.4 genomes in total. In contrast to BA.4, the prevalence of BA.5 increased more rapidly and on the 1st of June, 2022, 30.7% of all sequenced genomes from SARS-CoV-2 positive throat swabs were confirmed as BA.5 consensus sequences, with a total number of 1155 BA.5 cases [10]. Since BA.5 clearly was heading to replace BA.2 as the dominant SARS-CoV-2 lineage in Denmark, we investigated if our adapted RT-qPCR platform was able to detect Omicron BA.4 and BA.5 sub-variants as a group, since this RT-qPCR *per se* will not able to discriminate between them due to identical genomic sequences in the spike region.

WGS data obtained from Danish BA.4 and BA.5 sequences revealed that 92.3% of BA.4 consensus sequences (180/195) and 97.6% of BA.5 consensus sequences (323/331) carried the ΔH69/V70 mutation. The prevalence of the mutation 452R is 93% and 98.8% in BA.4 and BA.5, respectively, which confirmed that nearly all BA.4 and BA.5 carry both signature mutations. Based on alignment of unique sequences, 84 BA.4 and 51 BA.5, respectively, the primers and probes in our L452R RT-qPCR had a complete match with all BA.4 sequences. Of the 51 analyzed BA.5 sequences, one showed an A-to-T substitution in the middle of the L452R probe-binding region, whereas the primer-binding regions were conserved. The BA.5 genomes had a perfect match to the primers and probe in the degenerate ΔH69/V70 RT-qPCR whereas a single nucleotide mismatch, an A-to-G substitution, located in the middle of the ΔH69/V70 forward primer (ACATTCAACTC**G**GGACTTGTTCT), was observed in 10 of the 84 investigated BA.4 sequences (12%). This mutation did however not abolish the detection of the ΔH69/V70 mutation (Fig 4A).

Validation of the allelic ΔH69/V70-WT and the L452-WT discrimination RT-qPCRs were performed on 66 SARS-CoV-2 positive patient samples collected from 24th to 26th of May, 2022 at Danish community test facilities, covering a broad range of viral load with Ct-values between 16–33 in the initial E-Sarbeco RT-qPCR. Of these selected samples, 28 viral genomes were characterized as BA.4 and 38 as BA.5 by WGS. Results from our adapted Variant-PCR showed that all samples were double-positive for ΔH69/V70 and L452R, in accordance with either a BA.4 or BA.5 viral genotype (Fig 4).

Similar Ct-values were obtained in the ΔH69/V70, L452R and E-Sarbeco RT-qPCRs, indicating that all three RT-qPCRs perform equally well with BA.4 and BA.5 variants, confirming high specificity of the degenerate ΔH69/V70 probe for an Alpha VOC-like sequence, and

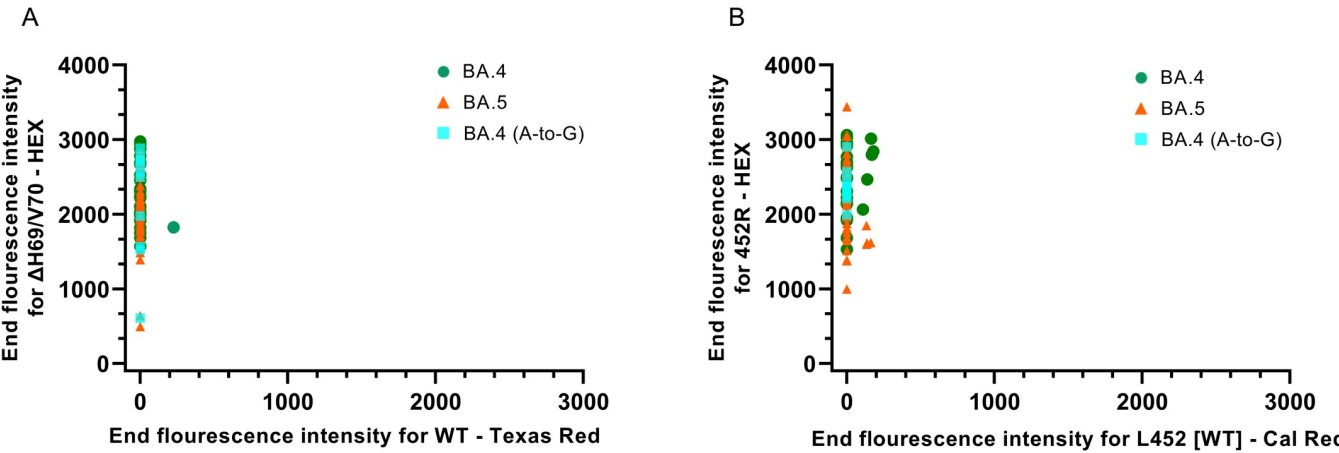

**Fig 4. Detection of ΔH69/V70 and L452R mutations in BA.4 and BA.5.** (A) Allelic discrimination analysis to differentiate between the ΔH69/V70 mutation and the WT sequence. (B) Allelic discrimination analysis to differentiate between the 452R mutation and the L452 WT sequence.

providing evidence for high specificity of this method for surveillance or large-scale studies. Furthermore, we observed no pronounced differences in amplification and detection of BA.4 viral genomes with a mismatch mutation in the sequence recognized by the ΔH69/V70 forward primer compared to patient samples with a conserved sequence (Fig 4).

In summary, our adapted Variant-PCR could rapidly detect Omicron BA.4 and BA.5, and distinguish them from earlier BA.1 and BA.2 variants, validated by comparison of RT-qPCR and WGS data from 66 patient samples analyzed in parallel.

## Discussion

Control of the pandemic caused by SARS-CoV-2 is challenged by the emergence of new variants, emphasizing the need for rapid and cost-effective detection methods that complement WGS in order to support control of transmission chains and assist public health response and decision-making. RT-qPCR is widely deployed in diagnostic virology [17]. In case of a public health emergency, laboratories can rely on this robust technology to establish new diagnostic tests for screening within their routine services before commercial assays become available. RT-qPCR strategies are of special interest in resource lean countries to complement WGS, the golden standard for SARS-CoV-2 variant surveillance. However, mutations in the targeting primer and probe sequences of new emerging variants can lead to a failure of the existing RT-qPCRs.

Currently, most commercial kits for SARS-CoV-2 variant determination utilize a so-called S-gene target failure phenomenon [18], where a detected variant positive for the *N* and *ORF1ab* genes, but negative for *S* gene is interpreted as Omicron, due to the large number of mutations in the S gene of this particular variant. The BA.1 sub-lineage showed S-gene target failure in RT-qPCRs due to the presence of the ΔH69/V70 mutation, however, an S-gene target failure strategy may not be able to detect the Omicron BA.2 lineage, which is currently the dominant SARS-CoV-2 virus circulating worldwide, as BA.2 is missing this particular deletion.

Recently, several single one-step RT-qPCRs reported for the specific detection of Omicron in clinical and environmental samples have been developed [19–21]. As Omicron continues to spread and evolve into several lineages and sub-lineages, these methods suffer, as they are not able to differentiate between multiple Omicron sub-lineages, although they have been described as highly specific and sensitive for Omicron.

The allelic discrimination RT-qPCR uses four probes, targeting two signature mutations in parallel, and has a very high accuracy, enabling detection of known VOCs almost in real-time (within 24h) after sample collection. We provide evidence for simultaneous, large-scale detection of the L452R mutation and ΔH69/V70 deletion in the spike sequence, using a degenerate probe to detect and differentiate between circulating Omicron- and Delta variants. To the best of our knowledge, this is the first RT-qPCR platform for specific detection and discrimination of Omicron sub-variants using a degenerate probe. This strategy enables detection of a key deletion, ΔH69/V70, regardless of a single nucleotide substitution or not, and hence, increases the robustness of the RT-qPCR. The samples for large-scale performance evaluation were chosen to challenge the robustness of these RT-qPCRs since lineage replacement is expected to occur during a pandemic transition period; the BA.1 sub-variant and its signature mutation, ΔH69/V70 with a single nucleotide substitution, was replaced by BA.2 with a wildtype genotype at both targets, ΔH69/V70 and L452R. Later, the appearance of BA.4 and BA.5 variants was observed in an increasing number of patient samples being positive for both key mutations. In fact, Omicron being the predominant variant worldwide, negative results in allelic discrimination RT-qPCR, but positive in other SARS-CoV-2 methods targeting conserved elements of the SARS-CoV-2 genome such as E-Sarbeco, may indicate the emergence of novel potential VOCs with S-gene mutations in target regions hindering recognition by the primers and/or probes.

In summary, after adapting our Variant-PCR platform with a degenerate ΔH69/V70 probe, we were able to detect the four most common Omicron variants in circulation, BA.1, BA.2, BA.4 and BA.5, as well as the Delta VOC, by combined analysis of two different signature mutations in parallel, ΔH69/V70 and L452R, validated by a comparison of paired RT-qPCR and WGS data. Furthermore, this method enables us to differentiate between Omicron and Delta VOCs, between BA.1 and BA.2 sub-lineages, and between BA.2 and emerging BA.4/BA.5 sub-lineages. If SARS-CoV-2 continues to evolve very contagious VOCs with potential to escape humoral immune response, which may lead to a new wave of COVID-19 infection, monitoring and screening might have to last for a prolonged period of time, making the use of degenerate probes an attractive strategy to adapt existing RT-qPCRs for SARS-CoV-2 detection.

## Supporting information

**S1 Fig. SARS-CoV-2 alpha variant standard sample detected by the original and the degenerate probe.** (A) Dilution of the TWIST control standard (SARS-CoV-2 Alpha variant) to a concentration of $10^{-3}$ copies/μl detected by the original and the degenerate probe targeting the ΔH69/V70. (B) Dilution row of the TWIST control standard (SARS-CoV-2 Alpha variant) detected by the original and the degenerate probe targeting the ΔH69/V70.
(TIF)

**S1 File.**
(TXT)

## Acknowledgments

We would like to extend our gratitude to Halenur Bayhan Ari, Kristina Finneisen and Caroline Mølsted Benfeldt for their assistance and technical support. This work would not have been possible without the Danish COVID-19 Genome Consortium (DCGC): Kasper S. Andersen, Martin H. Andersen, Amalie Berg, Susanne R. Bielidt, Sebastian M. Dall, Erika Dvarionaite, Susan H. Hansen, Vibeke R. Jørgensen, Rasmus H. Kirkegaard, Wagma Saei, Trine B.

Nicolajsen, Stine K. Østergaard, Rasmus F. Brøndum, Martin Bøgsted, Katja Hose, Tomer Sagi, Miroslaw Pakanec, David Fuglsang-Damgaard, Mette Mølvadgaard, Henrik Krarup, Christina W. Svarrer, Mette T. Christiansen, Anna C. Ingham, Thor B. Johannesen, Martín Basterrechea, Berit Lilje, Kirsten Ellegaard, Povilas Matusevicius, Lars B. Christoffersen, Man-Hung E. Tang, Kim L. Ng, Sofie M. Edslev, Sharmin Baig, Ole H. Larsen, Kristian A. Skipper, Søren Vang, Kurt J. Handberg, Marc T. K. Nielsen, Carl M. Kobel, Camilla Andersen, Irene H. Tarpgaard, Svend Ellermann-Eriksen, José A. S. Castruita, Uffe V. Schneider, Nana G. Jacobsen, Christian Ø. Andersen, Martin S. Pedersen, Kristian Schønning, Nikolai Kirkby, Lene Nielsen, Line L. Nilsson, Martin B. Friis, Thomas Sundelin, Thomas A. Hansen, Marianne N. Skov, Thomas V. Sydenham, Xiaohui C. Nielsen, Christian H. Schouw, Anders Jensen, Ea S. Marmolin, John E. Coia & Dorte T. Andersen

## Author Contributions

**Conceptualization:** Randi Jessen, Nicolai Balle Larsen, Arieh Sierra Cohen, Vithiagaran Gunalan, Ellinor Marving, Alonzo Alfaro-Núñez, Anders Fomsgaard, Katja Spiess.

**Investigation:** Nicolai Balle Larsen, Arieh Sierra Cohen, Vithiagaran Gunalan.

**Methodology:** Randi Jessen, Line Nielsen, Nicolai Balle Larsen, Arieh Sierra Cohen, Vithiagaran Gunalan, Ellinor Marving, Alonzo Alfaro-Núñez, Charlotta Polacek, Anders Fomsgaard, Katja Spiess.

**Project administration:** Anders Fomsgaard.

**Supervision:** Katja Spiess.

**Validation:** Ellinor Marving.

**Writing – original draft:** Randi Jessen, Nicolai Balle Larsen, Arieh Sierra Cohen, Vithiagaran Gunalan, Ellinor Marving, Charlotta Polacek, Anders Fomsgaard, Katja Spiess.

**Writing – review & editing:** Randi Jessen, Line Nielsen, Nicolai Balle Larsen, Arieh Sierra Cohen, Vithiagaran Gunalan, Ellinor Marving, Alonzo Alfaro-Núñez, Charlotta Polacek, Anders Fomsgaard, Katja Spiess.

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
