## [Decision Letter · Decision Letter 0]

12 Aug 2022

PONE-D-22-19692A RT-qPCR system using a degenerate probe for specific identification and differentiation of SARS-CoV-2 Omicron (B.1.1.529) Variants of ConcernPLOS ONE

Dear Dr. Spiess,

Thank you for submitting your manuscript to PLOS ONE. After careful consideration, we feel that it has merit but does not fully meet PLOS ONE’s publication criteria as it currently stands. Therefore, we invite you to submit a revised version of the manuscript that addresses the points raised during the review process.

We look forward to receiving your revised manuscript.

Kind regards,

Ruslan Kalendar

Academic Editor

PLOS ONE

Journal Requirements:

Reviewers' comments:

Reviewer's Responses to Questions

**Comments to the Author**

1. Is the manuscript technically sound, and do the data support the conclusions?

Reviewer #1: Yes

Reviewer #2: Yes

2. Has the statistical analysis been performed appropriately and rigorously? 

Reviewer #1: N/A

Reviewer #2: N/A

3. Have the authors made all data underlying the findings in their manuscript fully available?

Reviewer #1: Yes

Reviewer #2: Yes

4. Is the manuscript presented in an intelligible fashion and written in standard English?

Reviewer #1: Yes

Reviewer #2: Yes

5. Review Comments to the Author

Reviewer #1: 

1.Titles of X and Y axis in figure 3 and figure4 are the same probe L452, please check carefully.

2.No source information of BA.4 and BA.5 in material and method (Samples and controls), author should support this information.

3.Is there any founding in this study?

Reviewer #2: 

Two separate tests targeting the ΔH69/V70 deletion and the L452R point mutation in the S-gene were developed and used with a previously developed assay (E-gene) for detecting and differentiating SARS-CoV-2 Alpha variant, Delta variant, and BA.1/2/4/5 sub-variants of the Omicron strains. The assay was validated with 745 positive clinical samples, and genotypes were confirmed by WGS.

Most strains in Denmark, as well as in most part of the globe, are Omicron variants now, and BA.4 and BA.5 are the most dominant subtypes, with trace number of BA.2 strains. It looks like the L452R wild-type/Mutant assay will provide sufficient information for detection, as it can differentiate BA.1/BA.2 from the most prevalent strains of BA.4/BA.5. The differentiation of BA.2 from BA.1/4/5 using ΔH69/V70 deletion assay does not seems to be necessary. Additionally, the ΔH69/V70 assay, L452R assay and the E-gene assay are separate individual reactions. Taking one out will reduce the number of reactions for a given sample.

The assays were well-validated with clinical samples, and compared to WGS results. Yet, there was no standard curve generated to determine PCR efficiencies, dynamic range of detection, and limit of detection. Table 2 provided data of 5 dilutions, but there was no mentioning of the original concentration, thus limit of detection in terms of copy-number cannot be assessed. PCR efficiency and correlation coefficient were not calculated either (By the way, “Concentration” in the table may have been mislabeled, and should be “Dilution factor”).

There are many multiplex qPCR developed for SARS-CoV-2 detection and differentiation (J Clin Microbiol. 2021 Jul 19;59(8):e0085921; PLoS One. 2022 Mar 21;17(3):e0265748; Transbound Emerg Dis. 2022 Feb 26. https://doi.org/10.1111/tbed.14497; etc.). If authors can add the E-gene (E-Sarbeco) target into the L452R assay will further reduce the number of reactions, and thus detection cost.

Authors have mentioned E848K assay (lines 142-144), but this assay was not mentioned in the rest of the manuscript, and no primer/probe information were given.

In Supplementary Fig. 1, the original probe and the degenerate probe may have been mislabeled/switched.

6. PLOS authors have the option to publish the peer review history of their article (what does this mean?). If published, this will include your full peer review and any attached files.

Reviewer #1: No

Reviewer #2: **Yes: **Jianfa Bai

---

## [Author Response · Author response to Decision Letter 0]

5 Sep 2022

Response to the reviewers

We thank the reviewers for their enthusiasm for our study and replied in a step-by-step response below: 

Reviewer #1: 

1.Titles of X- and Y axis in figure 3 and figure 4 are the same probe L452, please check carefully.

Response: We thank the reviewer for pointing it out and corrected the y-axis in Figs 3A and 4A, accordingly. 

2. No source information of BA.4 and BA.5 in material and method (Samples and controls), author should support this information.

Response: We submitted the GISAID accessions IDs corresponding to the WGS-confirmed SARS-CoV Delta and Omicron patient samples included in this study for a better overview of the sequence information including the one for BA.4 and BA.5. During this process, we noticed that a total of nine sequences (four BA.4 and five BA.5) were not uploaded to GISAID for unknown reasons. These samples were used in our sequence alignments only and not in the RT-qPCRs, hence their exclusion from our study had no effect on our conclusions. We have corrected the manuscript as follows (line 269 -279): 

WGS data obtained from Danish BA.4 and BA.5 sequences revealed that 92.3% of BA.4 consensus sequences (180/195) and 97.6% of BA.5 consensus sequences (323/331) carried the ∆H69/V70 mutation. The prevalence of the mutation 452R is 93% and 98.8% in BA.4 and BA.5, respectively, which confirmed that nearly all BA.4 and BA.5 carry both signature mutations. Based on alignment of unique sequences, 84 BA.4 and 51 BA.5, respectively, the primers and probes in our L452R RT-qPCR had a complete match with all BA.4 sequences. Of the 51 analyzed BA.5 sequences, one showed an A-to-T substitution in the middle of the L452R probe-binding region, whereas the primer-binding regions were conserved. The BA.5 genomes had a perfect match to the primers and probe in the degenerate ∆H69/V70 RT-qPCR whereas a single nucleotide mismatch, an A-to-G substitution, located in the middle of the ∆H69/V70 forward primer (ACATTCAACTCGGGACTTGTTCT), was observed in 10 of the 84 investigated BA.4 sequences (12%). This mutation did however not abolish the detection of the ∆H69/V70 mutation (Fig 4A). 

At the time point of this study, BA.4 and BA.5 were up-coming variants in Denmark, and not yet dominant, hence no cultivated BA.4 or BA.5 virus isolates for use as positive control were available. Instead, virus isolates of Alpha variant B.1.1.7 and Delta variant B.1.617.2 were used as positive controls for the ∆H69/V70 and L452R key mutations, respectively. We tried to clarify it in the material and method part (line 108-115).

3. Is there any founding in this study?

Response: The study was performed at the Statens Serum Institut, in collaboration with Test Center Denmark, which are part of the Danish National Surveillance System for SARS-CoV-2. Therefore, no external funding was received for this study. 

Reviewer #2: 

Two separate tests targeting the ΔH69/V70 deletion and the L452R point mutation in the S-gene were developed and used with a previously developed assay (E-gene) for detecting and differentiating SARS-CoV-2 Alpha variant, Delta variant, and BA.1/2/4/5 sub-variants of the Omicron strains. The assay was validated with 745 positive clinical samples, and genotypes were confirmed by WGS.

Most strains in Denmark, as well as in most part of the globe, are Omicron variants now, and BA.4 and BA.5 are the most dominant subtypes, with trace number of BA.2 strains. It looks like the L452R wild-type/Mutant assay will provide sufficient information for detection, as it can differentiate BA.1/BA.2 from the most prevalent strains of BA.4/BA.5. The differentiation of BA.2 from BA.1/4/5 using ΔH69/V70 deletion assay does not seems to be necessary. Additionally, the ΔH69/V70 assay, L452R assay and the E-gene assay are separate individual reactions. Taking one out will reduce the number of reactions for a given sample.

Response: The major aim of this study was to show that a degenerate probe can be incorporated into existing RT-qPCRs to rescue the PCR after a change of a SARS-CoV-2 variant and a potential primer and probe mismatch. However, we agree that a single PCR target could be enough to differentiate BA.1/BA.2 and BA.4/5, but we think that including an additional PCR target is of advantage as the 452R mutation has been detected in BA.1 and BA.2 variants. Thus, including ΔH69/V70 RT-qPCR into the screening helps to minimize false positive results. 

The assays were well-validated with clinical samples, and compared to WGS results. Yet, there was no standard curve generated to determine PCR efficiencies, dynamic range of detection, and limit of detection. Table 2 provided data of 5 dilutions, but there was no mentioning of the original concentration, thus limit of detection in terms of copy-number cannot be assessed. PCR efficiency and correlation coefficient were not calculated either (By the way, “Concentration” in the table may have been mislabeled, and should be “Dilution factor”).

Response: We appreciated the input from the reviewer and changed it to dilution factor in the table. The ΔH69/V70 PCR efficiency has been determined in a previously study for the original ΔH69/V70 primers and probe (https://doi.org/10.1101/2021.10.25.2126548). We made this point more clear in the manuscript and wrote (line 206-211): Moreover, the fluorescence intensity was moderately reduced for the degenerate probe compared to the original probe targeting the ∆H69/V70 present in the SARS-CoV-2 Alpha variant (TWIST control) (S1-A Fig). However, the sensitivity of the ∆H69/V70 RT-qPCR was unchanged including the degenerate ∆H69/V70 probe instead of the original probe, as shown by a dilution row of the TWIST standard (Alpha variant) (S1-B Fig). The sensitivity of the original primer and probe of the ∆H69/V70 RT-qPCR was determined in a previous study (https://doi.org/10.1101/2021.10.25.2126548). 

There are many multiplex qPCR developed for SARS-CoV-2 detection and differentiation (J Clin Microbiol. 2021 Jul 19;59(8):e0085921; PLoS One. 2022 Mar 21;17(3):e0265748; Transbound Emerg Dis. 2022 Feb 26. https://doi.org/10.1111/tbed.14497; etc.). If authors can add the E-gene (E-Sarbeco) target into the L452R assay will further reduce the number of reactions, and thus detection cost.

Response: In our large-scale screening setup, primary diagnosis of SARS-CoV-2 is performed in parallel with an E-Sarbeco singleplex assay. Positive samples are re-analyzed for variant typing in 384-well format (https://doi.org/10.1101/2021.10.25.2126548). In this setup we have chosen to repeat the original E-Sarbeco assay in singleplex as a quality control for the original diagnostic assay. A multiplex PCR of L452R and E-Sarbeco targets would require extensive, time-consuming validation, since even small differences in sensitivity, could affect diagnostics, given the large numbers of sample processed in the Danish testing system. However, in small-scale we think it would be a good idea to run these PCRs as multiplex PCR with the L452R and E-Sarbeco PCR in one run, but this was out of the aim of this study.

Authors have mentioned E848K assay (lines 142-144), but this assay was not mentioned in the rest of the manuscript, and no primer/probe information were given.

Response: We apologize and deleted this information from the method part, as this assay was not included into this study. 

In Supplementary Fig. 1, the original probe and the degenerate probe may have been mislabeled/switched.

Response: We thank the reviewer for pointing this mistake out, but we checked the Supplementary Fig. 1 carefully and we could not find the mistake. In the Supplementary Fig. 1A, the degenerate ∆H69/V70 probe shows a reduced fluorescence intensity on the TWIST Alpha variant standard (10-3 copies/µl) compared to the original ∆H69/V70 probe. In contrast, the degenerate probe shows a better fluorescence intensity detecting SARS-CoV-2 BA.1 variants from patient samples than the original probe as shown in Fig. 1 and Fig.2.

---

## [Editor Report · Decision Letter 1]

7 Sep 2022

A RT-qPCR system using a degenerate probe for specific identification and differentiation of SARS-CoV-2 Omicron (B.1.1.529) Variants of Concern

PONE-D-22-19692R1

Dear Dr. Spiess,

We’re pleased to inform you that your manuscript has been judged scientifically suitable for publication and will be formally accepted for publication once it meets all outstanding technical requirements.

Kind regards,

Ruslan Kalendar

Academic Editor

PLOS ONE

---

## [Editor Report · Acceptance letter]

12 Sep 2022

PONE-D-22-19692R1 

A RT-qPCR system using a degenerate probe for specific identification and differentiation of SARS-CoV-2 Omicron (B.1.1.529) Variants of Concern 

Dear Dr. Spiess:

I'm pleased to inform you that your manuscript has been deemed suitable for publication in PLOS ONE. Congratulations! Your manuscript is now with our production department. 

Kind regards, 

on behalf of

Professor Ruslan Kalendar 

Academic Editor

PLOS ONE